# Hybrid Approach to Colony-Forming Unit Counting Problem Using Multi-Loss U-Net Reformulation

**DOI:** 10.3390/s23198337

**Published:** 2023-10-09

**Authors:** Vilen Jumutc, Artjoms Suponenkovs, Andrey Bondarenko, Dmitrijs Bļizņuks, Alexey Lihachev

**Affiliations:** 1Institute of Smart Computer Technologies, Riga Technical University, LV-1048 Riga, Latvia; vilens.jumutcs@rtu.lv (V.J.); artjoms.suponenkovs@rtu.lv (A.S.); andrejs.bondarenko@rtu.lv (A.B.); 2Institute of Atomic Physics and Spectroscopy, University of Latvia, LV-1586 Riga, Latvia; aleksejs.lihacovs@lu.lv

**Keywords:** colony-forming unit, deep learning, U-Net, segmentation

## Abstract

Colony-Forming Unit (CFU) counting is a complex problem without a universal solution in biomedical and food safety domains. A multitude of sophisticated heuristics and segmentation-driven approaches have been proposed by researchers. However, U-Net remains the most frequently cited and used deep learning method in these domains. The latter approach provides a segmentation output map and requires an additional counting procedure to calculate unique segmented regions and detect microbial colonies. However, due to pixel-based targets, it tends to generate irrelevant artifacts or errant pixels, leading to inaccurate and mixed post-processing results. In response to these challenges, this paper proposes a novel hybrid counting approach, incorporating a multi-loss U-Net reformulation and a post-processing Petri dish localization algorithm. Firstly, a unique innovation lies in the multi-loss U-Net reformulation. An additional loss term is introduced in the bottleneck U-Net layer, focusing on the delivery of an auxiliary signal that indicates where to look for distinct CFUs. Secondly, the novel localization algorithm automatically incorporates an agar plate and its bezel into the CFU counting techniques. Finally, the proposition is further enhanced by the integration of a fully automated solution, which comprises a specially designed uniform Petri dish illumination system and a counting web application. The latter application directly receives images from the camera, processes them, and sends the segmentation results to the user. This feature provides an opportunity to correct the CFU counts, offering a feedback loop that contributes to the continued development of the deep learning model. Through extensive experimentation, the authors of this paper have found that all probed multi-loss U-Net architectures incorporated into the proposed hybrid approach consistently outperformed their single-loss counterparts, as well as other comparable models such as self-normalized density maps and YOLOv6, by at least 1% to 3% in mean absolute and symmetric mean absolute percentage errors. Further significant improvements were also reported through the means of the novel localization algorithm. This reaffirms the effectiveness of the proposed hybrid solution in addressing contemporary challenges of precise in vitro CFU counting.

## 1. Introduction

CFU detection and counting is typically used in microbiology or cell biology to measure the number of viable initial cells in a culture or sample [1]. It is particularly useful for assessing the viability of bacteria or cell colonies during the development of cultures in the laboratory. It involves isolating and counting the number of visible bacterial or cell colonies growing on a solid or semi-solid surface (agar plate) inoculated with the sample [2]. Some popular CFU-counting applications include assessing the viability of a bacterial culture following exposure to a toxic compound or measuring the number of cells in a tissue sample following cellular transfection [3].

For many decades, detecting and counting CFUs was considered solely the responsibility of human experts. Diverse lighting and environmental conditions have prevented successful automation using image analysis techniques. Furthermore, CFU segmentation seems to suffer from drifting image acquisition conditions, background noise, variability of backgrounds, bacteria types, possible shapes, and the textures of agar plates being collected under varying conditions [4]. Nevertheless, following the latest advances in CFU segmentation and counting using deep learning techniques and, more specifically, U-Net architectures [5,6,7], greater accuracy has been achieved in biomedical probing, drug testing, and food safety applications. Moreover, the deep learning approach [8], which treats the CFU segmentation as an image recognition process, has reached higher precision and accuracy levels leading to substantial improvements in CFU counting. In addition to different U-Net architectures, which have been employed to increase the accuracy of the aforementioned segmentation task, one can combine several pre- and post-processing techniques to eliminate image artifacts and focus on the results within the visible bounds of an agar plate [9]. While U-Net models can improve CFU segmentation by combining both feature extraction and classification tasks, different edge-detection and image-processing techniques and filters [10,11] can further reinforce the counting approach by smoothing exterior CFU edges and distilling the localization of an agar plate.

From a general standpoint, the U-Net architecture lies at the heart of image segmentation in the biomedical domain, which, in turn, can be seen as a highly complex and error-prone task requiring state-of-the-art techniques to achieve accurate results. Recent advances in deep learning have led to significant progress in this area, leading to the development of specialized convolutional neural networks (CNNs) [12,13] such as U-Nets, deep reinforcement learning (DRL) [14,15] models, and generative adversarial networks (GANs) [16,17], which can improve image resolution or augmentation pipelines. Many of these models are used to accurately label objects in images for a variety of biomedical applications such as object detection, object recognition, and image classification. In addition, these techniques are used to identify subtle differences among similar images and separate the different components in a biomedical scene or image. All of these approaches greatly reduce the need for manual intervention and make the image segmentation process more time-efficient and less labor-intensive.

This paper proposes a novel hybrid approach to the CFU counting problem. The main idea is grounded in the multi-loss and multi-layer U-Net training objective tailored to provide an auxiliary signal of where to look for distinct CFUs. Compared to existing U-Net approaches for biomedical image segmentation, like the one used in the KiTS (https://kits-challenge.org, accessed on 1 June 2023) challenge [7], this objective introduces an additional term in the bottom-most bottleneck U-Net layer, where the receptive field of an encoder pathway typically extracts the most high-level and coarse-grained visual features. The main challenge of existing U-Net models applied in the aforementioned biomedical domain often boils down to very conservative loss functions and limited domain-specific target adaptations. To tackle this problem, enhance the training objective, and attain the finest and sharpest levels of granularity, various loss functions such as the dice similarity coefficient and the cross-entropy are combined with the proposed counting-tailored objective. Finally, a novel Petri dish localization technique is employed. This technique identifies the Petri dish bezel and the corresponding reflection zone within an image and outputs a segmentation mask for an accurate and precise estimate of the CFUs growing in the reflection zone and closer to the center of an agar plate. The latter constitutes another innovative algorithm proposed in this paper.

Following recent advances in CFU counting, self-normalized density maps [18] should be recognized as a promising approach to counting tasks. One can also utilize object detection methods such as YOLOv6 [19] for counting CFUs in terms of detected bounding boxes. In [18], the authors proposed a novel method of combining the statistical properties of the density segmentation maps with the baseline $U^{2}$-Net model [20]. Although these approaches serve as a solid foundation for the future, we strongly believe in the use of better-suited loss functions and domain-specific remodeling of U-Nets to tackle this challenging and convoluted domain of biomedical image segmentation.

To highlight the superiority of the proposed approach, this paper provides a comparative and comprehensive analysis of all examined U-Net models with and without an improved multi-loss reformulation, as well as with and without the Petri dish localization technique. The improvements of the Petri dish localization algorithm are presented in the ablation study in Section 3.3, highlighting the importance of applying proper post-processing and artifact-removing techniques. The remainder of this paper is organized as follows. The proposed methods, along with the datasets (materials) and experimental setup, are discussed in Section 2, where two core improvements, along with a web application for CFU counting, are illustrated in detail. The experimental results are presented in Section 3. In Section 4, we discuss the results obtained, the advantages and disadvantages of the proposed approach, and future research areas. Finally, Section 5 concludes the paper.

## 2. Materials and Methods

This section introduces the hybrid CFU counting approach, which can be seen as a pipeline with several steps. The approach consists of two essential components. The first involves the CFU segmentation task, in which a novel, improved multi-loss U-Net learning objective is proposed. The second focuses on the Petri dish localization and detection task. It allows for more fine-grained counting of the expected CFUs within a valid surface only by identifying its bezel and corresponding reflection zone, where visible CFUs can be duplicated. The second major step drastically reduces the latent levels of noise, e.g., flipped pixels, segmented-out artifacts, and other pixel-based errors inherent to image segmentation tasks, where targets are defined on a per-pixel basis. In between these two primary pipeline steps, there are other post- and pre-processing procedures responsible for removing artifacts within a single detected CFU region, e.g., smoothing exterior CFU edges.

### 2.1. Multi-Loss U-Net

The section introduces the proposed reformulation of the well-known U-Net architecture, which incorporates a supplemental loss function term in the training objective. To provide an auxiliary signal indicating where to locate CFUs for the U-Net model, a second ground-truth segmentation output map $Y^{mid}$ is defined. It constitutes the result of a binary mapping $F\mapsto R^{2}$ of all CFU centroids from the input 2D space to the re-sized 64×64 output 2D space of the bottom-most bottleneck U-Net layer. In the case of two or more CFUs that map to the same 2D spatial cell, we still keep its value unchanged and activated (for the binary mapping, this means that $Y_{ij}^{mid}=1$). Examples of both ground-truth output maps (corresponding to the same image) are illustrated in Figure 1.

Depending on the number of down-sampling blocks (e.g., max-pooling or convolution layers with a stride greater than 1), a different number of 2D spatial features can be obtained in the middle layer. All of the probed U-Net models are restricted to 1024×1024 input spatial dimensions, 5 down-sampling blocks with a stride of 2, and a 64×64 2D spatial resolution in the bottom-most U-Net middle layer. Moreover, to align the number of model-dependent feature maps in the middle layer, an averaging procedure is applied across these maps to obtain a single feature map. Finally, to achieve better alignment between the aforementioned ground-truth segmentation map $Y^{mid}$ and the averaged feature map of the middle layer $X^{mid}$, the following loss function is introduced:(1)$$L_{mid}\left(X^{mid},Y^{mid}\right)=-\frac{∥X^{mid}\cdot Y^{mid}{∥}_{1,1}}{∥X^{mid}{∥}_{F}{∥Y^{mid}∥}_{F}},$$
where ${∥\cdot ∥}_{F}$ relates to the Frobenius norm, ${∥\cdot ∥}_{1,1}$ is the entry-wise matrix norm, and · denotes element-wise multiplication. The latter loss function promotes similarity between output maps, thus enforcing the extraction of meaningful features only around pertinent CFU-containing regions of the input 2D space.

To achieve the final training objective, the optimization objective in Equation (Equation 1) is combined with typical segmentation-tailored losses, such as the dice similarity coefficient $L_{DSC}$ and the cross-entropy $L_{CE}$, which are defined solely for the output layer $X^{out}$ and the ground-truth segmentation $Y^{out}$ maps that match the input dimensions:(2)$$\begin{array}{c} L\left(X^{mid},X^{out},Y^{mid},Y^{out}\right)=L_{mid}\left(X^{mid},Y^{mid}\right)\\ +L_{CE}\left(X^{out},Y^{out}\right)+L_{DSC}\left(X^{out},Y^{out}\right), \end{array}$$
where the last two loss terms are defined as follows:(3)$$\begin{array}{c} L_{CE}\left(X^{out},Y^{out}\right)=-\sum_{c=1}^{2}{∥Y_{c}^{out}\cdot \log\left(X_{c}^{out}\right)∥}_{1,1} \end{array}$$(4)$$\begin{array}{c} L_{DSC}\left(X^{out},Y^{out}\right)=1-\sum_{c=1}^{2}\frac{∥X_{c}^{out}\cdot Y_{c}^{out}{∥}_{1,1}}{∥X_{c}^{out}{∥}_{1,1}{∥Y_{c}^{out}∥}_{1,1}+\epsilon }, \end{array}$$
where *c* spans an index of the target segmentation class, which, in retrospect, comes down to the foreground and background classes only, and $\epsilon =10^{-5}$ is a normalization constant to prevent exploding or imploding gradients.

To summarize the research findings, the paper introduces a novel multi-loss and multi-layer U-Net training objective, which comprises the well-known loss functions from the literature [7] and a novel loss term that builds upon an additional auxiliary signal. Any coefficients of the individual loss terms are not optimized, as their values are constrained to the $\left[-1,1\right]$ interval and are well-balanced.

### 2.2. Petri Dish Localization

The aforementioned approach is used for the Petri dish localization task to perform the downstream removal of bacterial colony reflections on an agar plate. The proposed approach consists of three instrumental parts: image pre-processing, Petri dish localization, and the segmentation of the reflection zone (see Figure 2).

The bacterial colony image pre-processing task consists of two parts: edge segmentation and blurring. The blurring procedure removes some of the image noise and small artifacts [21]. Bacterial colony images can be smoothed using a Gaussian blur. The Gaussian blur filter [22] can be formulated using the Gaussian probability distribution function calculus in a two-dimensional space, as in Equation (Equation 5):(5)$$\begin{array}{c} G\left(x,y\right)=\frac{1}{2\pi \sigma^{2}}e^{-\frac{x^{2}+y^{2}}{2\sigma^{2}}}, \end{array}$$
where *x* and *y* are two-dimensional coordinates and σ is the standard deviation (affects the strength of smoothing: a higher standard deviation value means stronger smoothing). High-frequency component suppression (the removal of small details) is achieved using this procedure. In this way, blurring prepares bacterial colony images for further analysis and downstream segmentation tasks.

The next important step is edge-based segmentation, which allows for the detection of sharp transitions in brightness levels in CFU images. Edge-based segmentation can be performed using a Canny edge detector [11]. Moreover, this segmentation prepares images for the subsequent agar plate localization task.

The aforementioned Petri dish localization task holds immense significance, as its outcome contributes to enhancing the precision of automated bacterial colony counting techniques. An agar plate usually has the shape of a circle. Therefore, the proposed localization method is based on the general equation of a circle in Equation (Equation 6):(6)$$\begin{array}{c} {\left(x-x_{0}\right)}^{2}+{\left(y-y_{0}\right)}^{2}=r^{2}, \end{array}$$
where *x* and *y* are the two-dimensional coordinates of the circumference; $x_{0}$ and $y_{0}$ are the two-dimensional coordinates of the circumference’s center; and *r* is the circumference’s radius. Furthermore, the main aim of this method is to find the parameters $x_{0},y_{0},r$ that produce a circumference that closely matches an observed bezel. Furthermore, it may be possible to use the circle Hough transform [10] for this purpose. At the beginning of the Petri dish localization step, the length of the dish radius is not known. Therefore, the 3D Hough space [10] should be used for the circle Hough transform. The 3D Hough space is represented using an accumulator array $A(x_{0},y_{0},r)$.

The next stage involves the segmentation of two reflection zones: the glass reflection and the water reflection. Figure 3 shows a schematic flow diagram of the reflection zone detection technique. First, the edge of the Petri dish is detected. Next, the glass edge is detected and, finally, the water edge is localized. The latter approach enables performing a complete Petri dish localization task while reducing the influence of highly disrupting CFU reflections on the proposed hybrid counting approach. The outcome of the algorithm presented in Figure 3 for one random sample is demonstrated in Figure 4.

### 2.3. Hybrid Approach

The section presents the hybrid approach, which can be postulated as a sequence of algorithmic steps. A simplified version of the proposed approach is presented step by step in Algorithm 1.

The breakdown of Algorithm 1 can be outlined using several crucial steps, which should be preserved for successful application. The first step is the actual prediction of the segmentation output map $X^{out}$ using the proposed multi-loss U-Net model $P_{seg}$. The aim of the next step is to post-process the predicted CFU regions to prevent any artifacts within (such as holes) or at the border of these regions. The latter can be performed by finding convex hulls and filling the inner holes using the scikit-image (https://scikit-image.org/, accessed on 1 June 2023) software library. Then, the Petri dish localization step is defined by the function $P_{dish}$. The latter function generates a full segmentation mask that covers the area between the outer bezel of the agar plate and the water’s edge inside. The final step depends on whether a valid mask has been found using $P_{dish}$. If successful, it is necessary to count the CFUs situated completely within the Petri dish but outside the identified mask, and to halve the count of the colonies detected within the mask. Otherwise, it is necessary to proceed to the normal counting technique using $count(X^{out})$ on the initial post-processed segmentation map $X^{out}$.
**Algorithm 1:** Proposed hybrid CFU counting approach.
**Data**: input image $X^{i}$

**Result**: number *N* of CFUs
**1**$X^{out}=P_{seg}\left(X^{i}\right)$;//(predict a segmentation output map(**2**$X^{out}=P_{convex}\left(X^{out}\right)$; //(find convex hulls of segmented-out regions(**3**$X^{out}=P_{fill}\left(X^{out}\right)$;//(fill the inner hollow parts of segmented-out regions(**4**$D^{out}=P_{dish}\left(X^{i}\right)$; //(locate a Petri dish, find the extended bezel mask(**5****if** $D^{out}$ *not empty* **then**(
**6**   $X^{inner}=M\left(X^{out},D^{out}\right)$; //(mask everything but inner part of the Petri dish(**7**   $X^{bezel}=M\left(X^{out},D^{out}\right)$; //(mask everything but bezel of the Petri dish(**8**    $N=count\left(X^{inner}\right)+count\left(X^{bezel}\right)/2$
**9****else**(
**10**  $N=count(X^{out})$



### 2.4. CFU Counting Application

To enhance and strengthen the acceptance of the novel approach, the authors of this paper have developed a web application that hosts the hybrid approach and provides seamless access to it. Within one tap and a single shot from a smartphone in a lab, one can accurately estimate the CFU count within a localized agar plate. This way, the working process is significantly streamlined and the productivity of lab personnel is boosted. Figure 5 illustrates the user interface of this application. The web UI consists of several integral widgets, including one for uploading images (top left) and one for viewing the statistics on the distribution of CFU sizes (top right). The other controls include a slider with a threshold in pixels of the minimum considered area of the bacterial colony, as well as a correction submission option. Other widgets represent the outcome of the hybrid approach, such as an image with the bounding boxes of the CFUs (in red) or the agar plate (in blue), as well as the estimated (raw counting based on the output of step 3 in Algorithm 1) and corrected (using Algorithm 1 completely) CFU counts within the agar plate. The web application was developed using Gradio v.3.24.1 (https://www.gradio.app, accessed on 1 June 2023) framework [23].

### 2.5. Datasets

This study uses two different CFU datasets, one of which is publicly available. The Annotated Germs for Automated Recognition (AGAR, https://agar.neurosys.com, accessed on 1 June 2023) dataset [24] (from NeuroSYS LTD located in Wroclaw, Poland) is an image database of microbial colonies cultured on an agar plate. It contains 18,000 photos of five different microorganisms taken under diverse lighting conditions with two different cameras. Following the experimental setup of [18], only high-resolution images were taken. Moreover, the authors of this paper did not follow the setup of providing ground-truth density maps using Gaussian blurring of the provided CFU centroids but rather drew a circle up to one-tenth of the minimal height or width (of the corresponding bounding box) in a radius around it. The other proprietary dataset consists of CFU images with different types of bacteria cultivated on an agar plate for later segmentation and estimation of the number of initially present CFUs. This dataset was collected in-house by our project partner (“Laboratorija Auctoritas” LTD, located in Riga, Latvia) and represents a tremendous effort in per-pixel segmentation and manual annotation. A summary of these datasets is provided in Table 1, together with the corresponding characteristics. Image samples from both datasets are shown in Figure 6.

### 2.6. Experimental Setup

All models were tested under the same experimental setup, involving either a 5-fold cross-validation or a random split with fixed-size training, validation, and test sets. During the training stage, a fixed number of epochs was used, where for each epoch, the model’s performance on the validation set was monitored, and the best-performing model (determined by the MAE score in Equation (Equation 8) on the validation dataset) was saved as a checkpoint. In the case of the AGAR dataset, all higher-resolution training and validation images were merged, and then three random splits were taken, where 10% was used as a test set, 18% as a validation set, and 72% as a training set. For the proprietary CFU dataset, there were only 150 annotated images and 242 test images with CFU counts available. In this scenario, a 5-fold cross-validation split was performed across the 150 annotated images, and the best-performing model from the checkpoints was evaluated on the test set. Splits for all models were kept identical to ensure comparability across all approaches. No thresholding was applied to the minimal region sizes or to steps other than those outlined in Algorithm 1 for post-processing. All images were re-scaled to 1024×1024 input spatial dimensions (using bi-linear interpolation) to match the setup in [18], and the standard z-score normalization technique was applied, i.e., each RGB channel was normalized according to its channel-wise mean and standard deviation:(7)$$z=\frac{X-\mu }{\sigma }.$$

For all datasets, this study reports the mean absolute error (MAE, Equation (Equation 8)) and the symmetric mean absolute percentage error (sMAPE, Equation (Equation 9)) for the CFU counts across all test images (agar plates). Additionally, performance metrics are presented specifically for images with ground-truth CFU counts below 100, as originally conducted in [18], to preserve comparability across all approaches. Finally, results for the YOLOv6 model under similar experimental settings are reported for both datasets, together with the best confidence threshold found (0.2 for the CFU dataset and 0.16 for the AGAR dataset). For YOLOv6, the setup provided in [19] was followed. We established a hard-coded split once, fixing the training, validation, and test sets mentioned above for all datasets. The well-established region-counting algorithm from the scikit-image v.0.15.0 (https://scikit-image.org/, accessed on 1 June 2023) software library was used to keep track of all CFU counts. It was applied across the output segmentation maps to obtain the predicted $\tilde{n_{i}}$ CFU counts per image, whereas $n_{i}$ was obtained from the JSON annotation files for the AGAR dataset or provided by a human expert for the proprietary CFU dataset.
(8)$$MAE=\frac{100\%}{N}\sum_{i=1}^{N}\left|n_{i}-\tilde{n_{i}}\right|$$
(9)$$sMAPE=\frac{100\%}{N}\sum_{i=1}^{N}\frac{|n_{i}-\tilde{n_{i}}|}{|n_{i}+\tilde{n_{i}}|}$$

The maximum number of epochs for the CFU dataset was set to 200, whereas for the AGAR dataset, it was set to 15. As the main optimization objective for $X^{out}$, the sum of the Tversky [25] and cross-entropy losses was used for all the evaluated U-Net architectures. Such an architecture is referred to as **single-loss** if only one ground-truth segmentation map $Y^{out}$ was used. The Tversky loss was initialized with α=0.5 and β=0.5, which effectively reduced it to a smoothed version of the dice similarity coefficient. An architecture is referred to as **hybrid** if Algorithm 1 was used and $P_{seg}$ was provided by such an architecture. The Adam optimizer [26] was used for training all the models, and the initial learning rate was set to 0.001. During the training stage, the learning rate was gradually decreased using a time-based decay scheduler. All experiments were performed using the TensorFlow v.2.9.0 [27] framework (from Alphabet, Inc. located in Mountain View, CA, USA) and the MIScnn v.1.1.9 [28] library for the variations of the U-Net model. The YOLOv6 model and the corresponding experimental setup infrastructure were obtained from the corresponding repository (https://github.com/meituan/YOLOv6, accessed on 1 June 2023). The code, together with the Petri dish localization library, was containerized and run using the Docker v.20.10.24 [29] (from Docker, Inc. located in Palo Alto, CA, USA) infrastructure. An Nvidia GeForce 1080 Ti GPU card (from Nvidia, Inc. located in Santa Clara, CA, USA) with 11 GB of RAM was used for training all the models. Other hardware specifications were as follows: 18-core Xeon e5-2696v3 CPU (from Intel, Inc. located in Santa Clara, CA, USA), 128 GB of RAM at 2133 MHz, and Linux (Ubuntu 20.04 LTS) operating system (from Canonical Group Limited located in London, UK). The source code was dockerized and added to Docker Hub (https://hub.docker.com/r/jumutc/cfu_analysis, accessed on 1 June 2023). The code is freely available using the following command: docker pull jumutc/cfu_analysis. The code is distributed under the GPLv3 (https://www.gnu.org/licenses/gpl-3.0.html, accessed on 1 June 2023) license and does not include any of the referenced datasets.

## 3. Results

### 3.1. Main Results

Table 2 presents the experimental results (without any stratification by the ground-truth CFU counts) on the CFU dataset and Table 3 demonstrates the performance on the AGAR dataset. The average MAE (Equation (Equation 8)) and sMAPE (Equation (Equation 9)) scores are reported with standard deviations across CV folds or random splits. The authors of this paper experimented with five diverse U-Net architectures, along with the YOLOv6 model (we report scores only for a single split), to provide an insight into the possibilities and performance gains one can achieve using the proposed approach. The results are aggregated across single-loss, hybrid (ours), and multi-loss (ours) formulations for the U-Net++ [30], Plain U-Net [7], Residual U-Net [31], Dense U-Net [32], and MultiRes U-Net [33] architectures. The best scores achieved across all U-Net architectures are highlighted in bold.

According to Table 2 and Table 3, the proposed hybrid approach outperformed all the counterpart models and the object detection YOLOv6 model by a large margin. The best MAE score for the Hybrid Multi-Loss Residual U-Net architecture for the CFU dataset—**12.17 ± 0.22**—and the Hybrid Multi-Loss Plain U-Net for the AGAR dataset—**1.89 ± 0.04**—were at least 1% to 3% better (in absolute values) than their single-loss counterparts and the YOLOv6 model. Consistent performance gains (within an absolute value of 1%) were also observed for most architectures when a U-Net model was paired with Algorithm 1, as indicated by the **hybrid** prefix. This justifies the application and practical importance of the Petri dish localization approach described in Section 2.2. Apart from localization gains, the most significant performance boost can be attributed to the enhanced training objective, which is discussed in Section 3.3.

Figure 7 compares two predicted segmentation output maps from the Dense U-Net architectures for one of the CFU test set images. The actual CFU count was 100, the multi-loss reformulation without further rectification via Algorithm 1 detected 101 CFUs, and the architecture’s single-loss counterpart detected 140 CFUs. As can be seen, the proposed multi-loss reformulation was much more precise and accurate in terms of segmented CFU regions. There was not a single artifact (on the left side of the image) that was segmented, resulting in an almost ideal estimation of the CFU count, i.e., 101 colonies compared to the 100 human-counted ones. These results clearly indicate the importance of the proposed auxiliary training objective in Equation (Equation 1), which spatially aligns the bottom-most U-Net layer features to the CFU centroids. The overall feature guidance and sparsification played a significant role in delivering the improved results shown in Figure 7.

### 3.2. Stratified Results

Finally, the results obtained under the same experimental setup and using the same CFU and AGAR datasets are presented to showcase the effectiveness and performance gains of the proposed approach, particularly in cases where mistakes are more pronounced, i.e., in samples with fewer than 100 ground-truth CFUs. These results were collected using the same models and test sets mentioned in Table 2 and Table 3, with an additional step of sample filtering on top. In hindsight, these results allow for a direct comparison with those of [18], as they too restricted their focus to a subset of the entire AGAR dataset to only those samples.

When comparing the scores (in bold) in Table 4 to the results of the self-normalized density map [18] approach, a major improvement can be seen, as the best MAE achieved for the authors’ proposed method (we report only reference numbers from [18]) was 3.65 and the best sMAPE was 4.05%. It is necessary to underscore one U-Net architecture that was especially notable for the AGAR dataset: Plain U-Net. Even when using a single-loss approach, it achieved much lower MAE and sMAPE scores. This observation is in contrast to the much worse performance results (compared to other architectures) on the proprietary CFU dataset and may indicate that this architecture performs better on homogeneous datasets such as the AGAR dataset, while it lacks generalization capabilities on more difficult datasets such as CFU or where co-variate shift problems appear.

A major improvement can also be identified for the proprietary CFU dataset in Table 5. All models integrated with Algorithm 1, along with the promising multi-loss U-Net reformulation, consistently outperformed the baseline architectures. This improvement is firmly rooted in the hybrid approach.

### 3.3. Ablation Study

To justify the practical use of Algorithm 1 and its constituents, an ablation study was performed to dissect it into two major alternatives: counting everything uniformly inside of $X^{inner}\cup X^{bezel}$ and counting CFUs inside of $X^{inner}$ and $X^{bezel}$ separately, as described in Algorithm 1. The experimental setup and dataset cross-validation splits were identical to those in Section 2.6. Table 6 and Table 7 present the results obtained only for the multi-loss hybrid architectures.

As can be concluded from Table 6, the proposed hybrid approach outperformed other ablated variants by a clear margin in terms of the MAE and sMAPE scores. Additionally, Table 7 demonstrates that for some architectures on the AGAR dataset, the proposed fully hybrid approach slightly worsened the observed MAE and sMAPE scores. This can be attributed to the absence of small CFUs and corresponding reflections at the bezel. In such a clean setting, any additional CFU count manipulations can only deteriorate the metrics. Future research may focus on a better reformulation of Equation (Equation 1), as different datasets may require more specialized and intricate handling.

## 4. Discussion

This paper has discussed a novel hybrid approach to the CFU counting problem. The approach is rooted in the multi-loss and multi-layer U-Net training objective tailored to provide an auxiliary signal of where to look for distinct CFUs. By applying a novel training objective, which introduces an auxiliary term in the bottom-most bottleneck U-Net layer, the receptive field of an encoder pathway is enriched. Additionally, various loss functions, such as the dice similarity coefficient and cross-entropy, are combined with the novel counting-tailored objective to boost overall training convergence and consistency. Finally, a novel Petri dish localization technique is used. This technique identifies different edges within an agar plate and outputs a segmentation mask for an accurate and precise estimate of CFUs growing in the reflection zone and closer to the center of a Petri dish.

An empirical evaluation of the proposed approach on both proprietary and public datasets underpins the research findings. The overall performance of all tested U-Net models, enhanced by the innovative loss term, surpasses that of any other model. This leads to the conclusion that additional signal guidance is always beneficial. On the other hand, post-processing can be seen as another integral part contributing to the observed improvement, as indicated by the ablation study. Taken together, the different components of Algorithm 1 confer a distinct advantage to the proposed hybrid approach over all other existing solutions.

Compared to the previous studies by Graczyk et al. [18], the demonstrated hybrid approach delivers better performance metrics and is much simpler in terms of practical implementation, as it builds upon standard U-Net architectures. Compared to other existing U-Net architectures, as shown in [6,7,20], the proposed approach is built upon these foundations and offers a problem-tailored loss function specifically designed to tackle CFU counting endeavors. Anyone can take an existing state-of-the-art U-Net model, train it, and seamlessly integrate it into the CFU counting application by implementing Algorithm 1 without knowledge of its internal details.

Finally, this section summarizes the advantages and disadvantages of the proposed approach:The advantages of the proposed approach are as follows:Possibility of seamlessly improving any U-Net architecture;Enhanced post-processing techniques that enable improving the contours and regions of segmented-out CFUs;Petri dish localization algorithm that enables counting CFUs only on a valid agar plate.The disadvantages of the proposed approach are as follows:Multiple losses during the training time may require a weighting scheme;Increased post-processing time and consumed resources.

## 5. Conclusions

This paper has investigated a novel multi-loss and multi-layer U-Net reformulation, which focuses on transmitting an auxiliary signal of where to look for distinct CFUs while solving a ubiquitous CFU counting problem. An additional loss term has been proposed for the bottom-most bottleneck U-Net layer, where the architecture extracts the most high-level and coarse-grained visual features. This novel loss function combines an averaged feature map of the aforementioned middle layer and a second ground-truth segmentation indicator map of the CFU positions in the input space. An additional Petri dish localization technique (along with other post-processing steps) has been proposed to enhance the overall performance and counting efficiency of the proposed hybrid approach.

A comprehensive empirical evaluation of the proposed solution on two diverse CFU datasets supports the research findings. Experimental improvements of at least 1% to 3% in the MAE and sMAPE scores are reported and underpin the effectiveness and benefits of the proposed solution. Additionally, when comparing the proposed approach to the recently published approach in [18], it was demonstrated that some notable performance gains of up to 3% in the MAE score can be achieved while using a simplified problem statement and a basic Plain U-Net reformulation embedded into Algorithm 1. Potential future research areas may include investigations into applying similar principles for multi-scale U-Net architectures, such as the previously mentioned $U^{2}$-Net [20] or recently proposed Multi-Path U-Net [34] models. Other potential future research directions may focus on improving pre- and post-processing techniques by leveraging improved low-level image-processing libraries such as BoofCV (http://boofcv.org/, accessed on 1 June 2023) and optimizing certain TensorFlow data ingestion workflows.

## Figures and Tables

**Figure 1 sensors-23-08337-f001:**
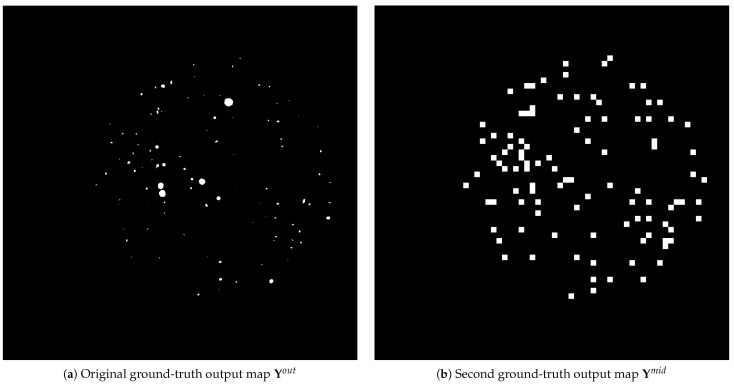
Examples of the original segmentation output map $Y^{out}$ and the one defined by the binary mapping $F\mapsto R^{2}$ as $Y^{mid}$.

**Figure 2 sensors-23-08337-f002:**
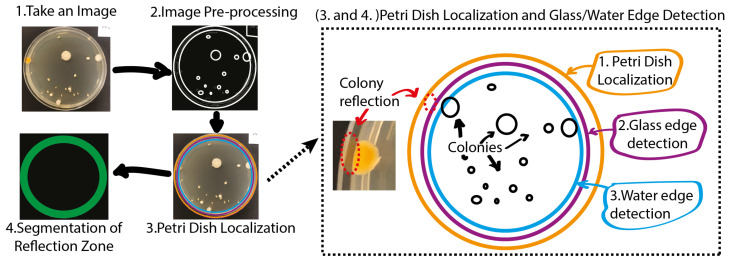
Image acquisition (1), image pre-processing (2), Petri dish localization (3), and segmentation of the reflection zone (4).

**Figure 3 sensors-23-08337-f003:**
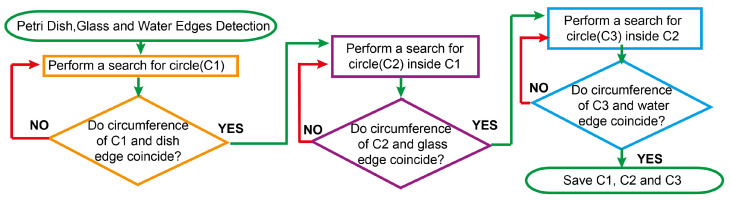
A schematic flow diagram of the underlying algorithm for the reflection zone detection technique. We would like to highlight several blocks, such as the search for C1—Petri dish edge; C2—glass edge; and C3—water edge. The final mask of the localized Petri dish bezel constitutes an area between C1 and C3, as can be observed in Figure 2 (3–4).

**Figure 4 sensors-23-08337-f004:**
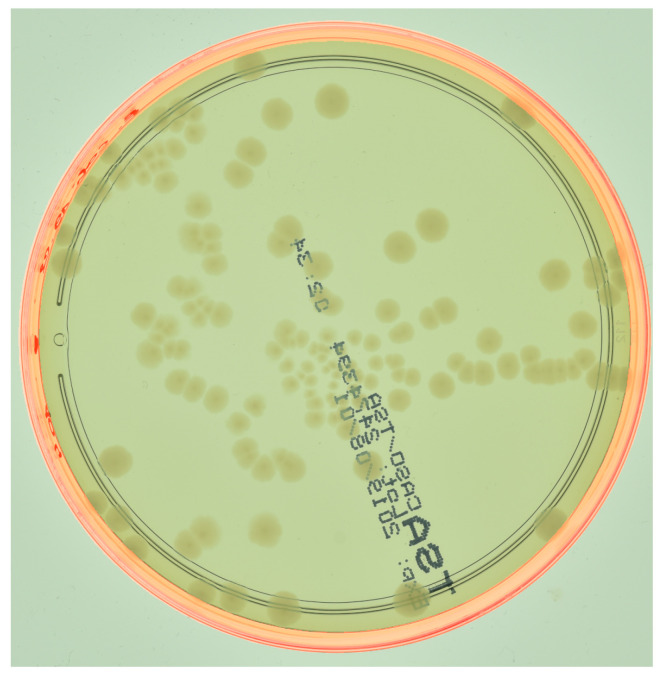
The bezel localization mask of an agar plate (from the AGAR dataset described in Section 2.5) is outlined in red. For visual clarity, the image is best viewed in color.

**Figure 5 sensors-23-08337-f005:**
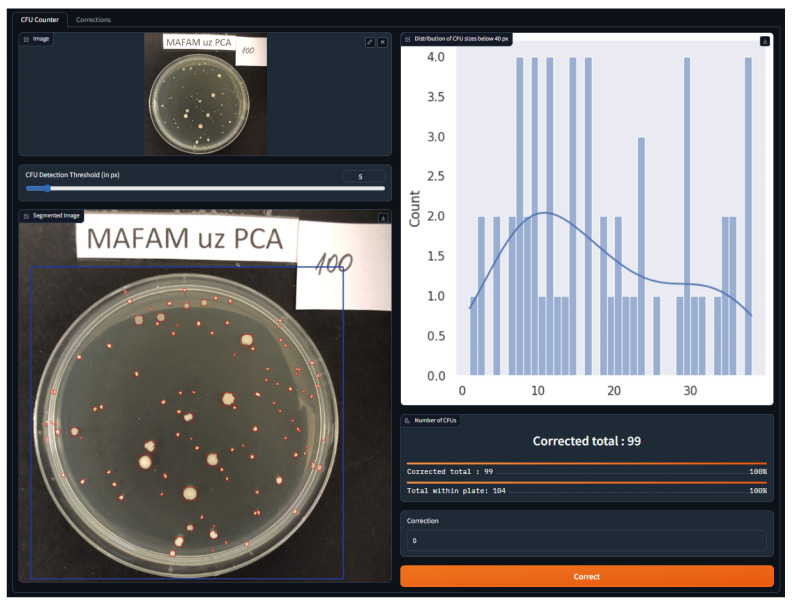
CFU- counting web application. The web UI is composed of different widgets, including one for uploading images (**top left**) and one for viewing the statistics on the distribution of CFU sizes (**top right**). The slider is used to control the minimum considered area in pixels of the bacterial colony. The submission input button is used to provide user feedback on the ground-truth CFU count. Other widgets represent the outcomes of our hybrid approach, such as the image with the bounding boxes of the CFUs (in red) or the agar plate (in blue), as well as estimated and corrected CFU counts. For visual clarity, better viewed in color.

**Figure 6 sensors-23-08337-f006:**
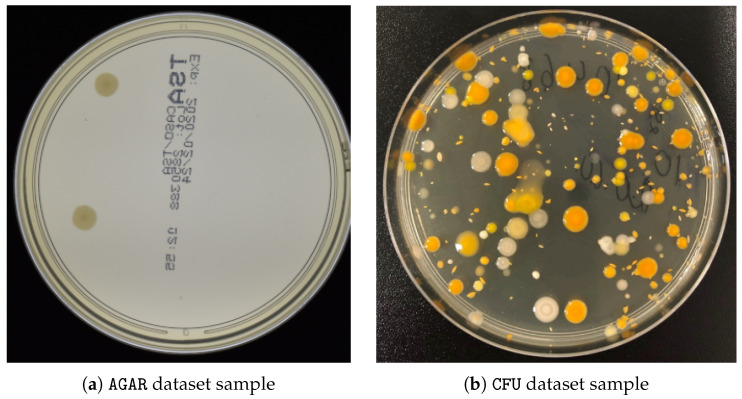
Magnified examples of AGAR (**a**) and proprietary CFU (**b**) datasets.

**Figure 7 sensors-23-08337-f007:**
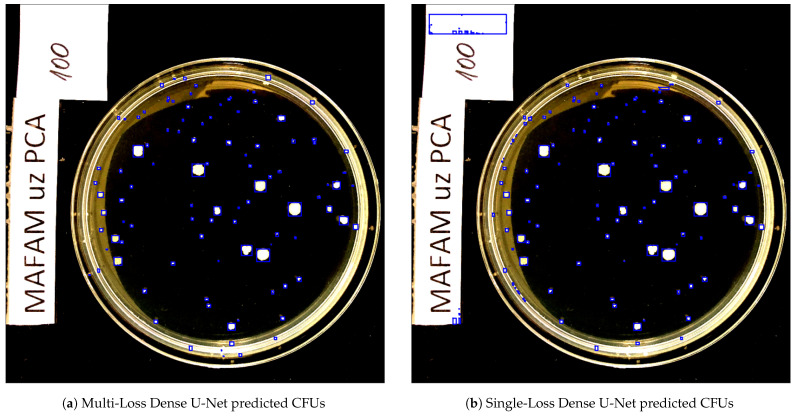
The predicted segmentation output for the CFU test set image from the Dense U-Net architectures, i.e., multi-loss versus single-loss. The results are represented by blue bounding boxes around the segmented CFUs. All images are shown in the normalized RGB spectrum, as described in Section 2.6. For visual clarity, the images are best viewed in color.

**Table 1 sensors-23-08337-t001:** Dataset characteristics. (AGAR high-resolution dataset, https://agar.neurosys.com, accessed on 1 June 2023).

Dataset	Number of Images	Image Sizes
AGAR high-resolution	6990	3632 × 4000 × 3
Proprietary CFU	392	3024 × 3024 × 3

**Table 2 sensors-23-08337-t002:** CFU dataset’s counting results for the YOLOv6, U-Net++, Plain U-Net, Dense U-Net, Residual U-Net, and MultiRes U-Net architectures. The results in bold (**hybrid multi-loss**) correspond to the complete hybrid solution in Algorithm 1, whereas the other rows correspond to different baseline U-Net architectures, either alone (**single-loss**) or embedded (**hybrid single-loss**) in Algorithm 1 instead of $P_{seg}$. YOLOv6 is provided for a complete reference.

Architecture	MAE	sMAPE
YOLOv6	19.69	12.41
Single-Loss U-Net++	18.54 ± 4.33	14.12 ± 3.51
Single-Loss Plain U-Net	40.44 ± 7.04	23.30 ± 3.05
Single-Loss Residual U-Net	21.24 ± 5.58	15.38 ± 5.04
Single-Loss Dense U-Net	19.14 ± 3.07	14.14 ± 2.36
Single-Loss MultiRes U-Net	15.14 ± 1.70	10.37 ± 2.21
Hybrid Single-Loss U-Net++ (ours)	15.93 ± 2.76	11.47 ± 2.41
Hybrid Single-Loss Plain U-Net (ours)	32.57 ± 3.43	19.93 ± 1.63
Hybrid Single-Loss Residual U-Net (ours)	18.51 ± 3.24	13.53 ± 3.88
Hybrid Single-Loss Dense U-Net (ours)	16.52 ± 2.26	12.17 ± 2.14
Hybrid Single-Loss MultiRes U-Net (ours)	14.13 ± 1.13	9.20 ± 1.61
Hybrid Multi-Loss U-Net++ (ours)	**13.06 ± 0.81**	**9.21 ± 0.76**
Hybrid Multi-Loss Plain U-Net (ours)	**24.72 ± 2.46**	**16.42 ± 1.49**
Hybrid Multi-Loss Residual U-Net (ours)	**12.17 ± 0.22**	**7.73 ± 0.35**
Hybrid Multi-Loss Dense U-Net (ours)	**14.10 ± 0.84**	**9.71 ± 0.68**
Hybrid Multi-Loss MultiRes U-Net (ours)	**13.42 ± 1.82**	**8.91 ± 1.85**

**Table 3 sensors-23-08337-t003:** AGAR dataset’s counting results for the YOLOv6, U-Net++, Plain U-Net, Dense U-Net, Residual U-Net, and MultiRes U-Net architectures. The results in bold (**hybrid multi-loss**) correspond to the complete hybrid solution in Algorithm 1, whereas the other rows correspond to different baseline U-Net architectures, either alone (**single-loss**) or embedded (**hybrid single-loss**) in Algorithm 1 instead of $P_{seg}$. YOLOv6 is provided for a complete reference.

Architecture	MAE	sMAPE
YOLOv6	7.93	12.28
Single-Loss U-Net++	17.70 ± 5.10	36.08 ± 17.39
Single-Loss Plain U-Net	3.05 ± 0.08	3.11 ± 0.22
Single-Loss Residual U-Net	11.93 ± 1.85	17.33 ± 3.83
Single-Loss Dense U-Net	17.74 ± 6.70	27.31 ± 9.84
Single-Loss MultiRes U-Net	7.55 ± 1.75	10.34 ± 2.48
Hybrid Single-Loss U-Net++ (ours)	17.76 ± 5.07	36.20 ± 17.43
Hybrid Single-Loss Plain U-Net (ours)	3.19 ± 0.07	3.37 ± 0.32
Hybrid Single-Loss Residual U-Net (ours)	11.57 ± 1.96	16.98 ± 3.82
Hybrid Single-Loss Dense U-Net (ours)	17.33 ± 6.44	27.48 ± 9.91
Hybrid Single-Loss MultiRes U-Net (ours)	6.91 ± 1.57	9.12 ± 2.18
Hybrid Multi-Loss U-Net++ (ours)	**10.82 ± 1.70**	**18.23 ± 4.30**
Hybrid Multi-Loss Plain U-Net (ours)	**1.89 ± 0.04**	**3.25 ± 0.15**
Hybrid Multi-Loss Residual U-Net (ours)	**7.67 ± 0.32**	**11.10 ± 1.56**
Hybrid Multi-Loss Dense U-Net (ours)	**6.78 ± 0.81**	**10.24 ± 0.57**
Hybrid Multi-Loss MultiRes U-Net (ours)	**6.64 ± 1.49**	**8.68 ± 1.78**

**Table 4 sensors-23-08337-t004:** AGAR dataset’s stratified (under 100 CFUs) counting results for the self-normalized density map, YOLOv6, U-Net++, Plain U-Net, Dense U-Net, Residual U-Net, and MultiRes U-Net architectures. The results in bold (**hybrid multi-loss**) correspond to the complete hybrid solution in Algorithm 1, whereas the other rows correspond to different baseline U-Net architectures, either alone (**single-loss**) or embedded (**hybrid single-loss**) in Algorithm 1 instead of $P_{seg}$. YOLOv6 and the self-normalized density map are provided for a complete reference.

Architecture	MAE	sMAPE
Self-Normalized Density Map	3.65	4.05
YOLOv6	4.27	11.71
Single-Loss U-Net++	10.29 ± 3.90	34.19 ± 17.42
Single-Loss Plain U-Net	1.23 ± 0.17	2.69 ± 0.23
Single-Loss Residual U-Net	6.68 ± 1.46	15.95 ± 3.78
Single-Loss Dense U-Net	11.78 ± 6.40	25.90 ± 10.12
Single-Loss MultiRes U-Net	3.84 ± 1.00	9.40 ± 2.34
Hybrid Single-Loss U-Net++ (ours)	10.35 ± 3.87	34.32 ± 17.46
Hybrid Single-Loss Plain U-Net (ours)	1.33 ± 0.17	2.95 ± 0.34
Hybrid Single-Loss Residual U-Net (ours)	6.15 ± 1.47	15.48 ± 3.69
Hybrid Single-Loss Dense U-Net (ours)	11.19 ± 6.15	25.97 ± 10.18
Hybrid Single-Loss MultiRes U-Net (ours)	3.06 ± 0.74	8.03 ± 1.98
Hybrid Multi-Loss U-Net++ (ours)	**5.81 ± 1.37**	**16.78 ± 4.29**
Hybrid Multi-Loss Plain U-Net (ours)	**0.88 ± 0.05**	**2.17 ± 0.13**
Hybrid Multi-Loss Residual U-Net (ours)	**3.61 ± 0.32**	**10.08 ± 1.72**
Hybrid Multi-Loss Dense U-Net (ours)	**3.30 ± 0.32**	**9.46 ± 0.44**
Hybrid Multi-Loss MultiRes U-Net (ours)	**2.90 ± 0.52**	**7.67 ± 1.48**

**Table 5 sensors-23-08337-t005:** CFU dataset’s stratified (under 100 CFUs) counting results for the YOLOv6, U-Net++, Plain U-Net, Dense U-Net, Residual U-Net, and MultiRes U-Net architectures. Most of the results in bold (**hybrid multi-loss**) correspond to the complete hybrid solution in Algorithm 1, whereas the other rows correspond to different baseline U-Net architectures, either alone (**single-loss**) or embedded (**hybrid single-loss**) in Algorithm 1 instead of $P_{seg}$. YOLOv6 is provided for a complete reference.

Architecture	MAE	sMAPE
YOLOv6	10.21	14.75
Single-Loss U-Net++	15.59 ± 4.94	19.11 ± 5.07
Single-Loss Plain U-Net	38.08 ± 6.22	23.30 ± 3.05
Single-Loss Residual U-Net	18.88 ± 8.46	20.79 ± 7.89
Single-Loss Dense U-Net	17.57 ± 3.81	19.28 ± 3.47
Single-Loss MultiRes U-Net	10.90 ± 2.68	13.25 ± 3.47
Hybrid Single-Loss U-Net++ (ours)	11.44 ± 2.85	14.98 ± 3.49
Hybrid Single-Loss Plain U-Net (ours)	30.84 ± 2.67	26.49 ± 2.20
Hybrid Single-Loss Residual U-Net (ours)	15.33 ± 6.32	18.10 ± 6.51
Hybrid Single-Loss Dense U-Net (ours)	14.30 ± 3.34	16.47 ± 3.32
Hybrid Single-Loss MultiRes U-Net (ours)	**9.04 ± 1.73**	11.39 ± 2.65
Hybrid Multi-Loss U-Net++ (ours)	**8.85 ± 0.90**	**11.87 ± 1.11**
Hybrid Multi-Loss Plain U-Net (ours)	**22.41 ± 2.34**	**21.89 ± 2.02**
Hybrid Multi-Loss Residual U-Net (ours)	**7.28 ± 0.35**	**9.37 ± 0.65**
Hybrid Multi-Loss Dense U-Net (ours)	**10.62 ± 1.15**	**9.71 ± 0.68**
Hybrid Multi-Loss MultiRes U-Net (ours)	9.11 ± 2.85	**11.24 ± 2.92**

**Table 6 sensors-23-08337-t006:** Ablation study results for the CFU dataset. **Multi-Loss only** indicates a U-Net architecture without any additional post-processing. **Multi-Loss (1)** indicates a multi-loss architecture where CFU counting is performed uniformly inside of $X^{inner}\cup X^{bezel}$. **Multi-Loss (2)** indicates the full approach outlined in Algorithm 1.

Architecture	Multi-Loss only		Multi-Loss (1)		Multi-Loss (2)	
	MAE	sMAPE	MAE	sMAPE	MAE	sMAPE
U-Net++	16.00 ± 1.90	13.01 ± 2.22	13.63 ± 0.87	9.93 ± 0.77	**13.06 ± 0.81**	**9.21 ± 0.76**
Plain U-Net	28.65 ± 2.79	18.61 ± 1.58	27.84 ± 2.83	18.02 ± 1.56	**24.72 ± 2.46**	**16.42 ± 1.49**
Residual U-Net+	12.87 ± 0.49	8.75 ± 0.59	12.75 ± 0.45	8.64 ± 0.54	**12.17 ± 0.22**	**7.73 ± 0.35**
Dense U-Net	15.73 ± 1.17	11.08 ± 0.83	15.36 ± 1.11	10.80 ± 0.79	**14.10 ± 0.84**	**9.71 ± 0.68**
MultiRes U-Net	14.90 ± 3.08	10.33 ± 2.65	14.44 ± 2.65	9.90 ± 2.25	**13.42 ± 1.82**	**8.91 ± 1.85**

**Table 7 sensors-23-08337-t007:** Ablation study results for the AGAR dataset. **Multi-Loss only** indicates a U-Net architecture without any additional post-processing. **Multi-Loss (1)** indicates a multi-loss architecture where CFU counting is performed uniformly inside of $X^{inner}\cup X^{bezel}$. **Multi-Loss (2)** indicates the full approach outlined in Algorithm 1.

Architecture	Multi-Loss Only		Multi-Loss (1)		Multi-Loss (2)	
	MAE	sMAPE	MAE	sMAPE	MAE	sMAPE
U-Net++	**10.72 ± 1.70**	**17.97 ± 4.21**	10.78 ± 1.70	18.09 ± 4.23	10.82 ± 1.70	18.22 ± 4.30
Plain U-Net	**1.74 ± 0.06**	**2.07 ± 0.15**	1.81 ± 0.05	2.20 ± 0.12	1.89 ± 0.04	2.35 ± 0.15
Residual U-Net+	7.76 ± 0.21	11.38 ± 1.39	7.72 ± 0.24	11.23 ± 1.57	**7.67 ± 0.32**	**11.10 ± 1.56**
Dense U-Net	7.13 ± 1.13	10.88 ± 0.91	6.92 ± 0.96	10.54 ± 0.74	**6.79 ± 0.81**	**10.24 ± 0.57**
MultiRes U-Net	**6.56 ± 1.48**	8.96 ± 1.69	6.60 ± 1.49	**8.63 ± 1.81**	6.64 ± 1.49	8.68 ± 1.78

## Data Availability

The AGAR dataset is available at https://agar.neurosys.com, accessed on 1 June 2023.

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
