# Peer review of "Hybrid Approach to Colony-Forming Unit Counting Problem Using Multi-Loss U-Net Reformulation"

_sensors, 2023, doi:10.3390/s23198337_

Round 1

Author Response

Hello,

Please consider the uploaded document as a complete reference/reply to all your comments and suggestions.

Many thanks in advance,

Dr.ir. Vilen Jumutc  

Reviewer 2 Report

Dear authors!

In my opinion, this paper is well-written and organized but has some obvious drawbacks. The contributions of this paper are a little bit ambiguous, and are not well summarized, Please add key contributions of your work in bullet form before the last paragraph of the introduction section.

The abstract can be written in a more professional way. The research findings should be added in this part. The numerical results, novelty of the work, and its applications can be presented and briefly discussed here.                       

The paper is based on very simple, textbook calculations and presents results that are obvious, well-known, and trivial. There is no novel theory, results or experiment and the reader expects far more significant and sophisticated content to qualify for a journal publication.

There are a few details regarding the overall simulation setup: I suggest considering a dataset with more number of images.

move the caption of Table 1 on the above table.

I suggest present date of tables 2-7 in the bar plot or some other suitable form to easily understand the results.

Increased post-processing time and consumed resources is the biggest challenge, How to overcome it? Please give future directions on it.

Add authors' contributions.

thanks

Author Response

Hello,

Please consider the attached document as a complete reference/reply to all your comments and suggestions.

Many thanks in advance,

Dr.ir. Vilen Jumutc  

Round 2

Reviewer 1 Report

The replied letter is not clear. The authors should listed out all the changes that has been made into the replied letter. Dont ask the reviewer to check again the manuscript.

Author Response

Dear sir,

Thank you very much for taking the time to review this manuscript. Please find the detailed responses in the attached document and the corresponding revisions are highlighted in red boxes to track changes in the re-submitted files.

Many thanks in advance!

Dr.ir. Vilen Jumutc
